# A Knowledge, Attitude, and Perception Study on Flu and COVID-19 Vaccination during the COVID-19 Pandemic: Multicentric Italian Survey Insights

**DOI:** 10.3390/vaccines10020142

**Published:** 2022-01-19

**Authors:** Cristina Genovese, Claudio Costantino, Anna Odone, Giuseppe Trimarchi, Vincenza La Fauci, Francesco Mazzitelli, Smeralda D’Amato, Raffaele Squeri

**Affiliations:** 1Department of Biomedical and Dental Sciences and Morphofunctional Imaging, Postgraduate Medical School of Preventive Medicine and Hygiene, University of Messina, 98121 Messina, Italy; crigenovese@unime.it (C.G.); giuseppe.trimarchi@unime.it (G.T.); vincenza.lafauci@unime.it (V.L.F.); francesco88xp@libero.it (F.M.); damato.esmeralda@libero.it (S.D.); 2Department of Health Promotion, Maternal and Infant Care, Internal Medicine and Excellence Specialties “G. D’Alessandro”, University of Palermo, 90127 Palermo, Italy; claudio.costantino01@unipa.it; 3Department of Public Health, Experimental and Forensic Medicine, University of Pavia, 27100 Pavia, Italy; anna.odone@unipv.it

**Keywords:** COVID-19, knowledge, attitudes, perception, vaccination, flu vaccination, COVID-19 vaccination

## Abstract

In January 2020, Chinese health authorities identified a novel coronavirus strain never before isolated in humans. It quickly spread across the world, and was eventually declared a pandemic, leading to about 310 million confirmed cases and to 5,497,113 deaths (data as of 11 January 2022). Influenza viruses affect millions of people during cold seasons, with high impacts, in terms of mortality and morbidity. Patients with comorbidities are at a higher risk of acquiring severe problems due to COVID-19 and the flu—infections that could impact their underlying clinical conditions. In the present study, knowledge, attitudes, and opinions of the general population regarding COVID-19 and influenza immunization were evaluated. A multicenter, web-based, cross-sectional study was conducted between 10 February and 12 July 2020, during the first wave of SARS-CoV-2 infections among the general population in Italy. A sample of 4116 questionnaires was collected at the end of the study period. Overall, 17.5% of respondents stated that it was unlikely that they would accept a future COVID-19 vaccine (*n* = 720). Reasons behind vaccine refusal/indecision were mainly a lack of trust in the vaccine (41.1%), the fear of side effects (23.4%), or a lack of perception of susceptibility to the disease (17.1%). More than 50% (53.8%; *n* = 2214) of the sample participants were willing to receive flu vaccinations in the forthcoming vaccination campaign, but only 28.2% of cases had received it at least once in the previous five seasons. A higher knowledge score about SARS-CoV-2/COVID-19 and at least one flu vaccination during previous influenza seasons were significantly associated with the intention to be vaccinated against COVID-19 and influenza. The continuous study of factors, determining vaccination acceptance and hesitancy, is fundamental in the current context, in regard to improve vaccination confidence and adherence rates against vaccine preventable diseases.

## 1. Introduction

In January 2020, the World Health Organization (WHO) reported that Chinese health authorities had identified a novel coronavirus strain never before isolated in humans: 2019-nCoV, which was then named severe acute respiratory syndrome from coronavirus-2 (SARS-CoV-2) [1].

The virus was associated with an outbreak of pneumonia cases, recorded as of 31 December 2019, in the central Chinese city of Wuhan, and it quickly spread around the world, leading to about 310 million confirmed cases since the start of the pandemic (data as of 11 January 2022) and to 5,497,113 deaths worldwide [2].

In Italy, the number of confirmed cases, as of January 2022, was 7,554,344, with 139,265 deaths attributable to COVID-19 infections [2]. 

Influenza virus usually affects millions of people during cold seasons, with high impacts in terms of mortality and morbidity; in Italy, there are about 7 million annual cases of flu and an average annual mortality excess rate ranging from 11.6 to 41 cases per 100,000 inhabitants. [3].

Some groups, such as the elderly and patients with comorbidities, are at a higher risk of acquiring severe respiratory diseases due to SARS-CoV-2 and influenza viruses, which could impact their underlying clinical conditions [4].

Furthermore, SARS-CoV-2 displays a clinical manifestation similar to influenza viruses, especially in the first phases of the disease, and this similarity could create an strain on the healthcare system during cold seasons [5,6].

Moreover, several studies have demonstrated a partial cross-protective role of flu vaccination [7,8].

In this context, the flu vaccination represents one of the most effective strategies that could be used to counteract the healthcare, social, and economic impacts of the influenza virus. At the beginning of the 2020/2021 cold season, it represented one of the best strategies to limit the impact of the COVID-19 pandemic [7,8].

In Italy, among the general population, a slight increase (16.7%) in vaccination coverage (CV) was recorded in the 2019/2020 flu season compared to the previous season (15.8%); in the elderly population (over 65 years of age), in the 2015/2016 season, a steadily increasing trend was recorded, reaching 54.6% during the 2019/2020 season [9].

However, this rate is much lower than the 75% threshold recommended by the World Health Organization (WHO), and for this reason, there is an urgent need to implement this vaccination in the target population [10]. This phenomenon could be, in part, attributable to opposition to vaccinations, known as vaccine hesitancy, which has been amplified in recent decades.

Vaccine hesitancy refers to a delay in the acceptance or the refusal of vaccines despite the availability of effective vaccines, and it depends on factors such as complacency, convenience, and confidence [11,12,13,14].

In early December 2020, the WHO issued an emergency use listing (EUL) for the Pfizer COVID-19 vaccine (BNT162b2) [15].

The UK became the first country in the world to approve the Pfizer/BioNTech COVID-19 vaccine for emergency use, a decision that was also taken by the U.S. Food and Drug Administration on 11 December 2020 [16].

Other vaccines were later licensed in the EU across Europe, and are currently being used in vaccination campaigns that started on 27 December 2020 [17]. To date, approximately 9.5 billion doses have been administered worldwide, and 59.2% of the general population is fully immunized [18].

Although attitudes toward COVID-19 vaccinations have already been investigated in the general population, studies that have examined the impacts of the COVID-19 pandemic, regarding the knowledge and attitudes on both the influenza and COVID-19 vaccinations, are lacking.

In the present study, during the COVID-19 pandemic, a sample of the Italian general population was studied and analyzed with regard to their intentions on getting vaccinated, as well as the knowledge, attitudes, and practices against influenza and COVID-19.

## 2. Materials and Methods

A multicenter, web-based, cross-sectional study was conducted between 10 February and 12 July 2020, during the first wave of SARS-CoV-2 infections among the general population in Italy.

Due to a general and stringent lockdown that affected the whole country from 8 March to 15 May 2020, the authors chose a web-based approach to distribute their questionnaire.

Twenty-one collaborating units (University of Bari, Bologna, Cagliari, Catania, Catanzaro, Florence, Foggia, Genoa, L’Aquila, Milano—Statale, Milano—Vita-Salute San Raffaele, Modena e Reggio Emilia, Napoli—Federico II, Napoli—Vanvitelli, Palermo, Parma, Roma—Sapienza, Sassari, Siena, Torino, Udine and Verona) participated in the questionnaire and data collection.

An Anonymous questionnaire was administered to participants of both sexes, who were 18 years of age and older, and who were able to understand Italian (to give informed consent and to complete the questionnaire).

Standardized questionnaires were distributed using the following techniques:Computer-assisted personal interview (CAPI), in which data were collected by an interviewer during a face-to-face meeting with the interview.Computer-assisted web interviewing (CAWI), in which the questionnaire was self-administered by the study participants and collected via email.

In the CAPI method, the interviews were conducted by medical staff (doctors/physicians) and medical residents in the Public Health and Preventive Medicine department of all participating units. This could lead to an extra-sampling error, to the interviewer’s effect.

It was therefore asked to participants to specify the date of administration with the CAPI mode, while in the CAWI mode—the system directly generated the response. An “ad hoc” category for classifying the period of administration was used in accordance with the epidemic–geographic evolution and legislation of the infection in Italy (on 18 February, we detected the first case in Italy; on 23 February, creation of municipal red zones; on 1 March, the extension of the zones to the whole of Lombardy; and finally, on 9 March, the general lockdown).

All participants were informed about the methodology used to ensure the confidentiality of data; written informed consent was obtained in accordance with Italian privacy laws. The interviews were carried out in locations that had adequate privacy. In the CAWI method, a link was sent by the interviewers to the study participants so that they could complete the questionnaire.

Considering the size of the population of residents (i.e., of both sexes, in the Italian cities where the universities agreed to participate in the study—approximately 10 million inhabitants, about one-sixth of the Italian general population), 3950 questionnaires should have been collected, with a degree of error calculation of 1%, in order to obtain a representative sample. All subjects able to complete the survey were invited to participate.

The protocol of the study was approved by the Ethics Committee of the University Hospital “G. Martino” of Messina (prot. 19-20 del 10/02/2020), and the other Local Ethics Committee accepted it.

### 2.1. Questionnaire Design

A questionnaire divided into four main sections and consisting of 25 items was given to all participants.

A standard questionnaire, regarding risk perception of an infectious disease outbreak, developed by the Municipal Public Health Service Rotterdam, Rijnmond (GGD), together with the National Institute for Public Health and the Environment (RIVM) in the Netherlands used in the project “Effective Communication in Outbreak Management; development of an evidence-based tool for Europe”, was appropriately modified for use [19].

The main sections of the questionnaire were divided into:Sociodemographic factors: age, sex, profession, place of residence, and origin. Italian regions were categorized into northern, central, and southern Italian regions.Working characteristics: in relation to profession, the study population was divided into healthcare professionals (HCPs), medical and non-medical university students, and general population.Health literacy: to investigate the health literacy level of the sample, we used the Italian version of the European Health Literacy Survey Questionnaire, based on six items.Knowledge of the disease: the participants were asked to report the main symptoms of a SARS-CoV-2 infection, selecting all of the applicable items, including cough, pneumonia, fever, myalgia, and flu-like symptoms.Cognition and behavioral patterns: a set of 20 items designed to measure cognition and behavioral patterns towards the COVID-19 disease, risk factors, worries about contracting SARS-CoV-2 (how would you feel if you were to contract COVID-19 in the coming year?), perceptions of disease susceptibility (do you think that you can contract COVID-19 in the coming year if you do not take any preventive measures?), preventive measure knowledge and vaccine hesitancy (dependent variable). In the case of vaccine hesitancy, respondents were asked about their willingness to get vaccinated if a flu or COVID-19 vaccine was offered to them. The answer options were: “very likely”, “somewhat likely”, “not sure”, “somewhat unlikely”, and “very unlikely”. The responses were dichotomized into two categories (1 = “hesitant”, including very/somewhat unlikely/not sure/; 0 = “confident”, including very likely/somewhat likely).

Intentions to get vaccine against some of the most important infectious diseases (seasonal flu, measles/mumps, and rubella, meningitis, and COVID-19) were also analyzed.

### 2.2. Statistical Analysis

All responses to the questionnaire were collected and summarized in Excel format.

Regarding the qualitative data (sex, educational degree, residence, etc.), the absolute and relative frequency and the 95% confidence interval were calculated. Meanwhile, the quantitative characteristics (age, number of people living in the house, etc.) were summarized using the mean, median, maximum and minimum values, standard deviation, interquartile interval, and 95% confidence limits.

Moreover, a univariate logistic regression analysis was performed to identify factors associated with willingness to receive the influenza vaccination and willingness to receive the COVID-19 vaccination (independent variables).

Estimates were expressed as crude odds ratios (ORs). All factors with a *p*-value ≤ 0.20 in the univariate analysis were included in a multivariable logistic regression model and adjusted odds ratios (adj-ORs), with 95% confidence intervals (CIs), were reported. The descriptive and inferential statistical analyses were performed using R software.

## 3. Results

A sample of 4208 questionnaires was administered, with a 97.8% survey completion rate. At the end period, a total of 4116 questionnaires were collected, of which, 3208 (77.9%) were through CAPI and 22.1% through CAWI. The median age was 32.96 (SD ± 12.96); females accounted for 64.1% of the sample.

Students represented 41.5% of the sample (of these, 8.7% were non-medical students and 35.1% were medical students), and 32.5% of the remaining sample belonged to the general population. HCPs accounted for 26% of the participants (14.2% medical doctors, 4.6% nurses, 0.5% social and health workers, 2.1% administrative workers, and 4.5% belonged to another healthcare profession).

The health literacy scale showed an average score of 2.58 ± 0.6. The social and demographic characteristics of the sample are reported in Table 1.

In our study, 54% of the participants were worried about contracting COVID-19, and 32.9% of the sample had a higher perceived risk of acquiring SARS-CoV-2 than the general population.

Over 50% (53.8%; *n* = 2214) of the sample was willing to receive a flu vaccination in the forthcoming vaccination campaign, and only 28.2% of cases had received it at least once in the previous five seasons.

Concerning the participants’ opinions regarding a future COVID-19 vaccine, 76% of the sample indicated that they would like to get a vaccine, while 17.5% were doubtful.

The main reasons behind vaccine refusal/hesitancy were a lack of trust in the vaccine, a lack of perception of susceptibility to the disease or the fear of side effects.

In Table 2, data regarding intentions to vaccinate against some of the most important vaccine preventable diseases (VPDs), such as seasonal flu, measles/mumps and rubella, meningitis, and COVID-19, are reported.

Higher rates of intention to be vaccinated were observed against meningitis (90.8%) and MMR (89.1%), followed by COVID-19 (76%) and seasonal influenza (53.8%).

As reported in Table 3, regarding the transmission route, mortality, and preventive measures for the SARS-CoV-2 infection–COVID-19, the interviewees were aware that it represented a communicable disease (99.1%, *n* = 4078), it could be asymptomatic (85%, *n* = 3498), and that COVID-19 could be fatal (91.9%, *n* = 3782).

Moreover, 89.9% were aware that, at the time of the survey, no vaccines for its prevention were available, and that only good hygiene measures (hand hygiene, the use of face masks, respiratory hygiene measures, etc.) could reduce transmission (83%, *n* = 3418) (Table 3).

Finally, 89.6% (2884) of the sample knew that a flu infection could be fatal. The mean knowledge score about a SARS-CoV-2 infection and COVID-19 disease was 6.73 (SD ±2.60) on a ten-point scale (data not shown in table).

In Table 4 and Table 5, uni- and multivariable analyses of factors associated with the intention to receive a flu vaccination during the 2020/2021 influenza season and COVID-19 vaccination were reported.

Regarding the flu vaccination, healthcare professionals (adj-OR = 1.49; CI 95% = 1.09–1.78), a higher knowledge score about SARS-CoV-2/COVID-19 (adj-OR = 1.04; CI 95% = 1.02–1.05), and at least one vaccination against flu during the previous influenza season (adj-OR = 54.2; CI 95% = 35.6–79.5) were the most significant predictors of influenza vaccination adherence (Table 4).

Determinants significantly associated with the willingness to undertake COVID-19 vaccinations were: a younger age class (less than 30 years of age), a higher score of knowledge about SARS-CoV-2/COVID-19 (adj-OR = 1.03; CI 95% = 1.01–1.05), and at least one vaccination against the flu during previous influenza seasons (adj-OR = 3.27; CI 95% = 1.78–7.81) (Table 5).

## 4. Discussion

According to more recent data from the World Health Organization, currently, there are more than 100 vaccines in clinical development and about 200 in the preclinical phase [17].

In Italy, to date, more than 60% of the population has received one dose of a COVID-19 vaccine, and about 50% have completed the vaccination course [20].

Two main challenges should be considered: (1) the allocation of a limited number of vaccination doses to the many people and countries who want to receive them; (2) vaccine hesitancy, particularly in developed countries that rapidly began their vaccination campaign from December 2020. Therefore, it is just as important to understand behavioral obstacles to immunization as it is scientific and logistical obstacles. [21].

In our study concerning participants’ opinions regarding future COVID-19 vaccines, the results show that 17.5% were doubtful. The reasons behind vaccine refusal/indecision were mainly a lack of trust in the vaccine, a lack of perception of susceptibility to the disease, and the fear of side effects.

In a study carried out by Silva J, et al., a 22-item, anonymous questionnaire was administered to 237 students who voluntarily attended two joint University Health Services and influenza vaccination clinics in November 2020: 92% were very/somewhat likely to receive a COVID-19 vaccine and 50% stated they would receive a COVID-19 vaccine as soon as possible.

In this study, only 3% of participants stated that they would never receive a COVID-19 vaccine. The top three COVID-19 vaccine-related concerns that were reported were safety (37%), effectiveness (24%), and limited information (16%) [22].

In a survey conducted by Cavaliere et al., “experience” with COVID-19 was shown to change attitudes, possibly positively, toward immunization in pregnancy [23]. Willingness to receive other vaccinations, such as meningococcal and MMR chickenpox, rather than COVID-19 and influenza vaccines, was reported, which was a similar result to other studies [24]. On the other hand, we face a decline in immunization coverage for many diseases worldwide, so countries should utilize other approaches that limit the spread of COVID-19, while providing the opportunity for immunization [25,26].

These results show that more efforts are necessary to increase vaccine compliance in the general population. Currently, vaccine hesitancy is one of the top 10 health threats, as reported by the WHO in 2019 [27].

Today, during the COVID-19 pandemic, vaccine hesitancy is an even more challenging health threat, as it can compromise the effectiveness of any new potential vaccines at the population level.

In previous seasonal campaigns, in Italy, the flu vaccine coverage lagged far below “the values”, of at least 75% as a minimum target, and even further below the 95% optimal target. In fact, in the last few years, a steady decline in vaccination coverage was reported, reaching 54.6% in the 2019/2020 season for the elderly population and 16.8% coverage in the general population [9].

Additionally, in healthcare workers, vaccination coverage is lower than the threshold recommended by the national and international healthcare authorities and far lower than the recommended threshold for other vaccinations, such as HBV, tetanus, or measles [28].

However, the fear related to the spread of COVID-19 and the preventive measures adopted, such as social distancing, the use of masks, and continuous hand-washing, have meant an increase in the rate of flu vaccine uptake and a decrease in the spread of flu in the Italian population.

The data published by the Ministry of Health regarding influenza vaccinations for the 2020–2021 season showed a significant increase in coverage in the general population, which rose from 16.8% in the previous season to 23.7% in the most recent season. In the elderly, above all, starting from the 2015–2016 season, a constant increase in coverage was shown, which stood at 54.5% in the most recent season. The coverage of the elderly increased by 12% compared to the previous season, reaching 66.5%, which means that vaccination coverage is gradually approaching the minimum threshold of 75% [29].

Furthermore, thanks to the preventive measures implemented and to greater adherence to the target vaccination campaign, the incidence of flu/flu-like syndromes has decreased, reaching a peak of 2 cases per 1000 people, which is the lowest incidence rate in the last 20 years, since the data on the territory were collected [30].

In fact, in the first weeks of 2020, the incidence of influenza-like illness (ILI) increased progressively until it reached an epidemic peak in the fifth week of 2020 (from 27 January to 2 February, 2020), with a level equal to about 13 cases per 1000 people, and in 2018/2019, the peak incidence level, from 28 January to 3 February, 2019, was equal to 13.8 cases per 1000 people [30].

In our study, we investigated the intentions of the studied sample—including university students, healthcare workers, and the general population—on receiving the flu vaccination, COVID-19 vaccination, and other vaccinations, 

The results show that most of the sample was willing to receive the flu vaccination during the COVID-19 pandemic period.

This could be important in this period, because, as previously reported, SARS-CoV-2 and influenza viruses share common transmissions and routes of entry, and demonstrate largely overlapping clinical features.

In several studies, authors have documented a negative correlation between influenza vaccination coverage and COVID-19 mortality at the country level, and in some cases, receiving a flu vaccination at the onset of COVID-19 clinical symptoms or shortly thereafter was associated with improved health outcomes [7,8,31]. Some authors claimed that the induction of the innate immune response by such a late vaccination possibly had a role in the more rapid and efficient SARS-CoV-2 clearance. Furthermore, previously described negative correlations between regions with lower influenza vaccination coverage and all the outcomes related to COVID-19 (seroprevalence, hospitalizations with symptoms, ICU hospitalizations, and number of deaths attributable to SARS-CoV-2) were observed in an Italian study [7]

Therefore, even though, in general, the population presented a low flu vaccination intention rate, the pandemic may have increased the population’s willingness to be vaccinated.

On average, flu-like syndromes (ILI) affect 9% of the Italian population every year, with a minimum prevalence of 4%, observed in 2005–2006, and a maximum of 15%, recorded in 2017–2018. In the current flu season, an incidence of 3.5 cases per 1000 people was observed, which is also the result of the restrictive measures adopted [According to the Authors in [31,32,33]. 

In this context, risk perception may increase a participant’s willingness to get vaccinated, especially in some categories, as a higher self-perceived risk of contracting a disease is associated with better adherence to the vaccination scheme. An even greater proportion of people will likely receive COVID-19 vaccinations once they become available. However, we found that general concerns about the pandemic and risk perceptions were not significantly associated with participants’ intentions to be vaccinated against COVID-19.

In a study conducted in the USA, most respondents were willing to receive the vaccine for themselves (75%) or their children (73%) [34]. In a recent multicentric survey across six countries, caregivers were more likely to immunize their child against the flu, with an increase of 15.8%; factors predicting willingness to immunize included the child’s up-to-date vaccination status, the caregivers’ influenza vaccine history, and the level of concern the child had regarding COVID-19 [35].

Furthermore, digital information and digital vaccine communication initiatives play a key role, and active cooperation between the media and scientific professionals contributes toward counteracting disinformation and provides up-to-date information. These—and other digital tools—represent the future of information, and they will became important tools for all medical applications, from diagnosis to data management [36].

Other studies conducted in different countries showed variable results: one multicenter study showed that 71.5% of respondents were very (or somewhat) likely to get vaccinated against COVID-19 [37]; in an Italian study, 59% of the respondents reported that they would likely be vaccinated against COVID-19 [38]. A systematic review showed high variability in the number of hesitant people (from very low, such as in China, 2%, or the UK, 4%, to very high, such as in Turkey, 44%, or the Czech Republic, 43%); however, most of the countries studied would not reach the value necessary to hit the herd immunity threshold [39,40,41,42].

These studies show that vaccination hesitancy is a widespread problem worldwide, but also underline that it is fundamental to understand social, demographic, and psychological determinants for the success of current national immunization plans, in order to highlight both the main determinants of vaccine hesitancy (to understand which groups of the population are more likely to refuse the vaccine) and the main factors favoring vaccination compliance.

An Italian study suggested two major pathways between health engagement and intention to vaccinate against COVID-19: a direct path between health engagement and intention to vaccinate and an indirect one that is mediated by the general attitude towards vaccination [43]. However, other studies show that the level of concern and perceived vulnerability during the COVID-19 pandemic are drivers that encourage people to get vaccinated [44,45,46,47].

In our study, health literacy in the multivariate analysis did not influence the attitudes toward receiving the flu vaccine and COVID-19 vaccine. Many previous studies have shown that health literacy plays a role (i.e., as a driver) in encouraging individuals to get vaccinated, and it can be influenced by a few key factors, including country of residence, age, and the type of vaccine received [48,49]. However, the relationship between health literacy and vaccination remains unclear; the understanding of the impact of health literacy and patient empowerment is fundamental for the concrete implementation of preventive behaviors.

Adherence to anti-COVID-19 measures, including flu immunization, could involve a proxy of health literacy. In other studies, the ability to detect fake news and health literacy through education and communication programs was associated with higher vaccine acceptance.

This implies that current and future vaccination campaigns should primarily rely on increasing individual health literacy and correct communication of the role of the vaccine, its efficacy and safety, risks of non-vaccination, and the social and economic consequences of a lack of immunization of the population for some infectious diseases. Only through citizen empowerment, correct communication, and adequate healthcare is it possible to achieve cultural change directed towards a committed and conscious approach toward self-health management, to ensure adequate levels of immunization against the risk of infectious diseases.

In interpreting the results of our survey, we acknowledge its main limitations: sampling was opportunistic, so we cannot generalize the results to the entire Italian population; the perception of risk was compared with age, but not with the presence of comorbidities, which is an important factor in the clinical manifestation and prognostic implications of the disease.

Additionally, the observational nature of the study poses problems related to the presence of systematic errors, such as selection mechanisms in the recruitment of study participants (selection bias), selective recall, or inconsistent data collection, measurement errors, the presence of confounding factors, social desirability bias, etc.

We used the six-item European Health Literacy Survey Questionnaire (HLS-EU-Q6) for convenience due to the time to complete the survey, but we know that it was not found to be valid, in a study from France [50].

The survey was conducted in the early months of the pandemic when vaccines and effective treatments had not been developed. In our study, all participants were recruited at different times. Table 5 displays how their scores changed over time; this could have led to a distortion bias, but our sample size was high, and so it was reduced.

A limit of this study is that we did not track the participants longitudinally to check their vaccination statuses after the survey administration, as well as their perceptions and their thoughts on boosters and children’s vaccines. Additionally, we did not investigate the number of participants who received flu vaccines in 2020.

Another limit is the different administrations of the survey that could impact the type of answer and introduce extra-sampling errors, in particular, on the effects of the interviewer.

Nonetheless, to our knowledge, this is the first study that investigated the impact of the COVID-19 pandemic on the intentions of flu and COVID-19 vaccinations, and possible influencing factors among a population.

## 5. Conclusions

In conclusion, in the current context, the factors determining vaccination hesitancy are fundamental, as they allow us to target population groups that could endanger a society’s goal of achieving herd immunity, which, at present, unfortunately seems to be a distant goal.

Furthermore, some studies have reported a decrease in mortality related to COVID-19 alongside higher flu vaccine uptake rates, but other studies are needed to determine whether there is any causally-associated link between the higher uptake of flu vaccination rates and a lower number of deaths caused by COVID-19. 

It would be of value to standardize a COVID-19 vaccine hesitancy questionnaire regarding language and statistical evaluation. The creation of such a tool would be an important contribution to this field of research.

Vaccine hesitancy, in regard to confidence, complacency, convenience, etc., is complex; we did not investigate all factors, including cultural, socioeconomic, native vs. immigrant status, pregnancy status, etc., but this article sheds light on several factors, such as complexities.

Finally, the COVID-19 pandemic has increased the acceptance rate—in unvaccinated people—of flu vaccinations, reminding us that, in an era in which non-communicable diseases consume most of the economic resources of healthcare systems, we must be prepared to deal with emerging and reemerging infectious diseases.

## Figures and Tables

**Table 1 vaccines-10-00142-t001:** Sociodemographic and working characteristics of the study sample (*n* = 4116).

	N	%
Gender		
*Male*	1474	35.9
*Female*	2637	64.1
Mean age ± SD	32.96	±12.96
Area of Origin		
*Northern Italy*	1361	33.1
*Center Italy*	397	9.6
*Southern Italy*	2358	57.3
Category		
*Healthcare professional*	1037	26
*Student*	1708	41.5
*General population*	1371	32.5
Influenza vaccination status in the last 5 influenza seasons		
*Never*	2948	71.7
*At least once*	1162	28.3
Health literacy (scale from 0 to 4)	mean score: 2.58 ± 0.6 SD median: 2.5 (0.83 IQR)	
Concern about contracting COVID-19	2214	54
Perceived risk of acquiring SARS-CoV-2		
*Equal/lower than general population*	2760	67.1
*Higher than general population*	1356	32.9

**Table 2 vaccines-10-00142-t002:** Intention to get vaccinated against some common vaccine preventable diseases (VPDs).

	No	Yes % (*n*)	I Do Not Know
Seasonal flu 2020/2021	33.1 (1363)	53.8 (2214)	12.9 (532)
Measles, mumps, rubella	6.7 (275)	89.1 (3666)	4.1 (168)
Meningitis	3.5 (143)	90.8 (3739)	5.6 (229)
COVID-19 vaccination	6.5 (266)	76 (3128)	17.5 (722)

**Table 3 vaccines-10-00142-t003:** Knowledge of the study sample regarding SARS-CoV-2 infection-COVID-19 disease.

	Incorrect Answer/ Answer Not Given	Correct Answer
SARS-CoV-2 infection is a communicable disease	0.9 (37)	99.1 (4078)
SARS-CoV-2 infection is always symptomatic	92.8 (3819)	7.1 (294)
COVID-19 can be deadly	8.1 (333)	91.9 (3782)
There is a vaccine against COVID-19	98.3 (4046)	1.7 (69)
SARS-CoV-2 infection can only be prevented by good hygiene measures	16.9 (697)	83 (3418)

**Table 4 vaccines-10-00142-t004:** Factors associated with influenza vaccination adherence in the univariable (crude OR) and multivariable (adj-OR) analyses. (adj-OR: adjusted odds ratio; 95% CI: confidence intervals 95%).

Variable	Willingness to Receive Influenza Vaccination (Yes vs. No)			
	Crude OR (95% CIs)	*p*-Value	adj-OR (95% CIs)	*p*-Value
Gender				
*- Female*	Reference	0.77		
*- Male*	1.02 (0.89–1.15)			
Age classes				
*- ≤30 years*	Reference	<0.001	Reference	0.18
*- ≥31 years*	0.72 (0.64–0.82)		0.86 (0.68–1.07)	
Area of Origin				
*- Northern Italy*	Reference	0.42		
*- Center Italy*	1.05 (0.72–1.25)			
*-* *Southern Italy*	0.89 (0.64–1.22)			
Category				
*- General population*	Reference	<0.01	Reference	<0.05
*- Students*	1.27 (1.05–1.67)		1.17 (0.96–1.52)	
*- Healthcare professionals*	1.59 (1.15–1.89)		1.49 (1.09–1.78)	
Concern about contracting SARS-CoV-2	1.19 (1.07–1.26)	<0.01	0.97 (0.88–1.16)	0.55
Perceived risk of acquiring SARS-CoV-2				
*- Equal/lower than general population*	Reference	0.91		
*- Higher than general population*	1.02 (0.85–1.22)			
Parents of children under 12 years of age				
*- No*	Reference	<0.01	Reference	0.47
*- Yes*	0.63 (0.53–0.75)		0.78 (0.70–1.19)	
Knowledge about SARS-CoV-2/COVID-19 score	1.06 (1.05–1.07)	<0.001	1.04 (1.02–1.05)	<0.001
Health literacy score	1.38 (1.26–1.51)	<0.001	1.13 (0.98–1.31)	0.09
Influenza vaccination during previous influenza seasons				
*- No*	Reference	<0.001	Reference	<0.001
*- Yes*	47.7 (34.5–66.0)		54.2 (35.6–79.5)	

**Table 5 vaccines-10-00142-t005:** Factors associated with willingness to accept COVID-19 vaccination in the univariable (crude OR) and multivariable (adj-OR) analyses. (adj-OR: adjusted odds ratio; 95% CI: confidence intervals 95%).

Variable	Willingness to Receive COVID-19 Vaccination (Yes vs. No)			
	Crude OR (95% CIs)	*p*-Value	adj-OR (95% CIs)	*p*-Value
Gender				
*- Female*	Reference	0.23		
*- Male*	1.09 (0.94–1.27)			
Age classes				
*- ≤30 years*	Reference	<0.001	Reference	<0.001
*- ≥31 years*	0.50 (0.43–0.57)		0.63 (0.50–0.85)	
Area of Origin				
*- Northern Italy*	Reference	0.27		
*- Center Italy*	1.15 (0.88–1.32)			
*-* *Southern Italy*	0.79 (0.61–1.44)			
Category				
*- General population*	Reference	0.12	Reference	0.13
*- Students*	1.28 (0.85–1.72)		1.17 (0.88–1.72)	
*- Healthcare professionals*	1.12 (0.64–1.32)		1.09 (0.81–1.31)	
Concern about contracting SARS-CoV-2	0.99 (0.97–1.02)	0.89		
Perceived risk of acquiring SARS-CoV-2				
*- Equal/lower than general population*	Reference	0.06	Reference	0.69
*- Higher than general population*	1.06 (0.99–1.13)		1.02 (0.95–1.12)	
Parents of children under 12 years of age				
*- No*	Reference	<0.01	Reference	0.19
*- Yes*	0.59 (0.49–0.72)		0.87 (0.69–1.09)	
Knowledge about SARS-CoV-2/COVID-19 score	1.04 (1.03–1.06)	<0.001	1.03 (1.01–1.05)	<0.01
Health literacy score (continuous variable)	1.14 (1.03–1.26)	<0.001	1.00 (0.88–1.15)	0.93
Influenza vaccination during previous influenza seasons				
*- No*	Reference	<0.001	Reference	<0.001
*- Yes*	3.26 (2.67–3.99)		3.27 (1.78–7.81)	

## Data Availability

Data available on request due to restrictions of privacy.

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
