# Peer review of "A Knowledge, Attitude, and Perception Study on Flu and COVID-19 Vaccination during the COVID-19 Pandemic: Multicentric Italian Survey Insights"

_vaccines, 2022, doi:10.3390/vaccines10020142_

Round 1

Reviewer 1 Report

Dear Authors,

I have perused your manuscript titled Knowledge, attitudes and perpection (sic) on flu and COVID-19 vaccination during the COVID-19 Pandemic: a multicentre Italian Survey insights (sic).
Overall, the manuscript is potentially valuable in terms of its stated objective, but suffers from poor structuring and form. The language is shoddy from the very beginning, starting with the title. In the abstract and following sections, things do not get much better (e.g.: line 34:  "In the present study, intentions to get vaccinated, knowledge and attitudes of Italian general population against influenza and COVID-19 during the COVID-19 pandemic." What does it even mean? Main verb is missing).
The issue of vaccine hesitancy is quite multifaceted and complex, and the article, for all its flaws, sheds some light on several such complexities with solid enough methodology. The way in which the findings are interpreted and articulated could be improved, and broadened in scope by a more thorough analysis of other contributing factors that may impact hesitancy rates: cultural, socioeconomic, native vs immigrant status, pregnant women, etc..., in addition to the dynamics determining changes in attitude over time, as the pandemic unfolds.
Studies have in fact been focused on the key elements underpinning vaccine hesitancy, so I would advice you to take a look at the following sources in order to broaden your perspective and the paper discussion and compare data:

Silva J, Bratberg J, Lemay V. COVID-19 and influenza vaccine hesitancy among college students. J Am Pharm Assoc (2003). 2021 Nov-Dec;61(6):709-714.e1. 

Cavaliere AF, Zaami S, Pallottini M, Perelli F, Vidiri A, Marinelli E, Straface G, Signore F, Scambia G, Marchi L. Flu and Tdap Maternal Immunization Hesitancy in Times of COVID-19: An Italian Survey on Multiethnic Sample. Vaccines (Basel). 2021 Sep 29;9(10):1107.

Fridman A, Gershon R, Gneezy A. COVID-19 and vaccine hesitancy: A longitudinal study. PLoS One. 2021 Apr 16;16(4):e0250123. doi: 10.1371/journal.pone.0250123. PMID: 33861765; PMCID: PMC8051771.

Wang Q, Xiu S, Zhao S, Wang J, Han Y, Dong S, Huang J, Cui T, Yang L, Shi N, Liu M, Han Y, Wang X, Shen Y, Chen E, Lu B, Jin H, Lin L. Vaccine Hesitancy: COVID-19 and Influenza Vaccine Willingness among Parents in Wuxi, China-A Cross-Sectional Study. Vaccines (Basel). 2021 Apr 1;9(4):342. 

Readability needs to be improved and the manuscript must be proof-read by a native speaker of English. 
Errors such as "2 hundreds" (line 237) are rife throughout the manuscript, as are poor vocabulary/grammar choices.

In addition, the references are not compliant with MDPI standards, they need to be done over as in:

Fridman, A.; Gershon, R.; Gneezy, A. COVID-19 and Vaccine Hesitancy: A Longitudinal Study. PLoS One 202116, e0250123, doi:10.1371/journal.pone.0250123.

I feel this praiseworthy contribution on your part needs to be submitted again in a majorly improved, nimbler and more readable version, and with a deeper interpretation of data and findings.

Sincerely,

Author Response

Dear thanks to the opportunity to review my manuscript. 

Reviewer 2 Report

Review of Knowledge, attitudes and perception on flu and COVID-19 vaccination during the COVID-19 Pandemic: a multicentre Italian Survey insights

  1. Overview. In this paper, the authors report the results of a survey given to understand attitudes surrounding influenza and COVID-19 vaccination acceptance. The determinants of those attitudes were then described. Receiving an influenza vaccination in the prior year was the strongest predictor of the intent to accept a COVID-19 vaccination when/if available.
  2. Syntax and grammar
    1. Line 2: perpectionperception
    2. Lines 31-34: I’d change to two sentences
    3. Lines 34-35: do not make sense. It seems to be a fragment of a larger thought.
    4. Lines 38-40: Instead consider, “Overall, 17.5% of respondents stated they were unlikely to accept a future COVID-19 vaccine.”
    5. At this point, I won’t continue a line-by-line review of syntax and grammar. I suggest working with an English language editor.
  3. Comments
    1. Lines 40-41: I would include the percent of respondents for each of these reasons
    2. Line 43: the conclusion
    3. Line 113: how were the patients recruited? Enrolled?
    4. Lines 124-125: In the legend for Table 1, the Ethics Committee approval to conduct this study is noted. I would place that in the text instead. As this is a multi-center study, did the Ethics Committees of the other participating centers independently approve the protocol or is there some sort of reciprocity amongst Ethics Committees whereby the primary study site reviews and approves and the other sites accept that review and then grant approval? Also, are the surveys anonymous or deidentified for the data analysis?
    5. I would like to see the actual survey tools used in this study. It can be placed in an appendix or supplementary material. In this way, Table 1 is not needed as the actual survey will show the reader the language used, the possible responses, and the scoring rubric.
    6. Line 149: I am concerned about the health literacy tool. The 6 item European Health Literacy Survey Questionnaire HLS-EU-Q6 was not found to be valid in this study from France: Rouquette A, Nadot T, Labitrie P, Van den Broucke S, Mancini J, Rigal L, etal. (2018) Validity and measurement invariance across sex, age and eduation level of the French short versions of the European Health Literacy Survery Questionnaire
    7. Line 167: I think statistical methodology could be described in more detail. Specifically, the construct of the regression models and the handling of the variables.
    8. Line 181: how many people were invited to participate? In other words, are the 4116 collected surveys from a larger cohort of people eligible and invited to participate? Or were there exactly 4116 participants with 100% survey completion rate? Also, eligibility criteria are not clear.
    9. Table 2
      1. I would like to see the IQR for age. Were there any participants 65 years or older?
      2. Row: worry about getting COVID-19 disease is not formatted properly. Also, to what does the mean score refer? I personally find it hard to believe that just over half of those surveyed were worried about getting COVID-19. I am in the USA and thought that Italy was severely affected by COVID-19. And the survey was conducted in the early months of the pandemic when there was no hope for a vaccine or effective treatment in sight. So, I would have expected the vast majority of participants were worried about getting infected.
  • Row: perceived risk of acquiring SARS-CoV-2 is not formatted properly.
  1. Table 3 is a surprise—I didn’t realize vaccination intent for measles, mumps, rubella and meningitis was also being assessed.
  2. Why are tables 5 and 6 formatted differently? And it is not clear what the adjusted model is. Also, this is not the typical way uni- and multi-variable regression tables are presented. Variables in rows and the different models tested as columns. The data in the cells are then the regression coefficients or odds ratios with measures of significance.
    1. Is health literacy treated as a continuous or a categorical variable?
  3. Discussion
    1. Lines 283-287: the incidence of influenza like illness was slightly lower in the first weeks of 2020 compared to 2018/2019 flu season?
  4. Overall impression: This paper provides limited new information. A better study would have been tracking these 4116 participants longitudinally to check on vaccination status, check if their perceptions changed, check thoughts on booster, child vaccines. Also, the number of participants who received a flu vaccine in 2020 should be reported. From a structural standpoint, table formatting is inconsistent meaning it takes extra time and effort to understand the information. Moreover, papers evaluating COVID-19 vaccine hesitancy are becoming commonplace. It would be of value to standardize a COVID-19 vaccine hesitancy questionnaire in regard to language and statistical evaluation. Presenting such a survey tool would be and important contribution.

Author Response

Dear thank you to the opportunity to review my manuscript 

Round 2

Reviewer 1 Report

Dear Authors,

I believe you have been largely successful in improving the overall quality of your article, in terms of quality of presentation and balance.

Approval for publication should therefore be granted, in light of the article's significance and research value.

Sincerely.

Author Response

We thank you for the opportunity to make the major revision requested by reviewer 2 and we are glad that reviewer 1 appreciated our first round of review

Reviewer 2 Report

Reviewer 2,

Review of Knowledge, attitudes and perception on flu and COVID-19 vaccination during the COVID-19 Pandemic: a multicentre Italian Survey insights

My comments are in response to revised manuscript and authors’ responses to specific comments.

I thank the authors for their thoughtful revisions of their manuscript and for their responses to specific comments.

I have ongoing concerns regarding methodology and presentation of the data.

  1. I am concerned that some participants had a personal interview (CAPI) while for others the surveys were self-administered (CAWI).
    1. We need to know how many respondents were in each group
    2. I suspect that the participants receiving a personal interview with a health care professional responded differently than they would have if they completed the survey on their own.
  2. I still don’t know how many people actually completed the surveys: 4025 (97.8% of 4116)?
  3. Line 212, over 50%, not almost
  4. Line 222, what is meant by interviewed population? What is the comparison population?
  5. Line 230, you don’t need the second regarding
  6. Line 239, Tables 4 and 5?
  7. Tables 3, 4, 5: I would left justify the first column. And I would right justify the data columns.
  8. Tables 4, 5: I still don’t know what the fully adjusted model is
  9. Table 6: data is unclear. Does the number refer to the number of surveys completed during that regulatory time period?
    1. If so, that adds up to 3240 participants.
    2. I would add the calendar dates that correspond to the regulatory time periods
    3. So, you are not comparing the same group of respondents at each regulatory period. These are different people each time and may have different characteristics, so statistical comparison may be biased.
  10. Line 391: uptake isn’t quite accurate as the authors didn’t record actual vaccination, just the respondents’ attitude towards receiving the vaccine.

Author Response

We thank you for the opportunity to make the major revision requested by reviewer 2 and we are glad that reviewer 1 appreciated our first round of review

This manuscript is a resubmission of an earlier submission. The following is a list of the peer review reports and author responses from that submission.